# A Reference Laboratory Surveillance on Fungal Isolates from Patients with Haematological Malignancy in Japan

**DOI:** 10.3390/jof7100806

**Published:** 2021-09-27

**Authors:** Yutaro Hino, Akira Watanabe, Rio Seki, Shokichi Tsukamoto, Yusuke Takeda, Emiko Sakaida, Katsuhiko Kamei

**Affiliations:** 1Medical Mycology Research Center, Division of Clinical Research, Chiba University, Chiba 260-8673, Japan; yutarohino@chiba-u.jp (Y.H.); rseki@chiba-u.jp (R.S.); k.kamei@faculty.chiba-u.jp (K.K.); 2Department of Hematology, Chiba University Hospital, Chiba 260-8673, Japan; shokichi.tsukamoto@chiba-u.jp (S.T.); take-you@hospital.chiba-u.jp (Y.T.); esakaida@faculty.chiba-u.jp (E.S.)

**Keywords:** invasive fungal disease, haematological disorder, immunocompromised host, *Aspergillus*, *Candida*, *Fusarium*

## Abstract

Invasive fungal disease (IFD) in patients with haematological disorders is a fatal disease, making rapid identification and treatment crucial. However, the identification of the causative fungus is often difficult, sometimes even impossible. There have been few reports concerning the causative species of IFD. This study aimed to investigate the epidemiology and causative organism of IFD in patients with haematological diseases in Japan. We analyzed the IFD cases among the patients with haematological malignancies identified at the Medical Mycological Research Center, Chiba University, between 2013 and 2019. The most common underlying disease was acute myeloid leukaemia (34.3%). Forty-six point one percent of IFD patients received haematopoietic stem cell transplantation (HSCT). The major pathogens were *Aspergillus*, *Candida*, and *Fusarium. Aspergillus fumigatus* was the most common *Aspergillus* species, and *Candida fermentati* and *Fusarium petroliphilum* were the most common *Candida* and *Fusarium* species, respectively, in this analysis. Furthermore, various cryptic species and non-*albicans Candida* were identified. The drug susceptibility of such relatively rare strains suggests that analysis of the causative fungi should provide valuable information for therapeutic options. Therefore, our study indicated that it is clinically significant to identify the organism in as much detail as possible.

## 1. Introduction

Despite improved diagnostic methods and novel antifungal agents, invasive fungal diseases (IFD) in patients with haematological diseases are still fatal [1]. Patients with acute myeloid leukaemia (AML) and myelodysplastic syndrome (MDS) are known to be at high risk for IFD due to prolonged neutropenia [2,3]. Furthermore, IFD is one of the most severe prognostic complications of hematopoietic stem cell transplantation (HSCT), with a reported incidence of 5–10% [4,5,6,7]. For this reason, some guidelines recommend consideration of antifungal prophylaxis during intensive chemotherapy for acute leukaemia or during HSCT [8,9].

According to previous reports, the most common causative agents of IFD are *Candida* and *Aspergillus* [5], but the identification of fungi is often more complicated than that of bacteria, and in many cases is not easy. The prognosis of IFD is poor, and therefore it is clinically important to identify the organism quickly so that it can be treated as soon as possible. However, due to its rarity, little is known about the causative agents of IFD in Japan. Our previous study identified the causative organism of invasive fusariosis in Japanese patients with haematological disorders [10,11], but other organisms have not yet been investigated. The Medical Mycological Research Center (MMRC), Chiba University, is designated as an “Advanced Progressive Laboratory for Infectious Diseases” by The Japanese Association for Infectious Diseases and The Japanese Society for Clinical Microbiology and identifies fungal isolates from various domestic medical facilities as a public reference centre. In this study, we analyzed the causative fungi of IFD in patients with haematological diseases in Japan using our MMRC database.

## 2. Methods

### 2.1. Epidemiological Study of IFD in Patients with Haematological Disorders in Japan

We analyzed IFD among the domestic patients with haematological malignancies identified in our centre between 2013 and 2019. This is a retrospective study of nationwide data concerning etiological agents, deposited to the MMRC (provided through the National Bio-Resource Project [NBRP], Japan http://www.nbrp.jp/, accessed on 21 June 2021). In addition to analysis of the causative organism, clinical information regarding the organism was reviewed. Those with insufficient clinical information were excluded (Figure 1). The study population and exclusion are shown in Figure 1. A total of 102 cases of IFD and the relevant clinical information of each patient were retrieved from the MMRC database. All IFD cases were classified as proven or probable IFD according to the European Organization for Research and Treatment of Cancer/Invasive Fungal Infections Cooperative Group and the National Institute of Allergy and Infectious Diseases Mycoses Study Group (EORTC/MSG) criteria [12]. The clinical information included baseline demographical characteristics, age, underlying diseases, and the history of HSCT. The study was conducted according to the Declaration of Helsinki and approved on 18 June 2021 by the Research Ethics Committee of Medical Mycology Research Center, Chiba University under a number MMRC-REC 21–25.

### 2.2. Fungal Identification

All clinical isolates were cultured on a potato dextrose agar (PDA) slant at 25 °C for 3–7 days. The DNA extraction was performed with PrepMan Ultra Sample Preparation Reagent (Thermo Fisher Scientific, Warrington, Cheshire, UK) according to the manufacturer’s instructions. Subsequently, species-level identification was performed with the nuclear ribosomal internal transcribed spacer (ITS) and D1/D2 region gene sequencing, as previously described [10]. Briefly, we amplified the nuclear ribosomal ITS region, including ITS1, 5, 8S, and ITS2 by PCR with the primers ITS5 and ITS4. We performed Sanger sequencing with the BigDye^®^ Terminator v3.1 Cycle Sequencing Kit (Thermo Fisher Scientific, Tokyo, Japan) and the ABI 3130xl Genetic Analyzer (Thermo Fisher Scientific, Tokyo, Japan) according to the manufacturer’s instructions. We also performed basic local alignment search tool (BLAST) searches of the sequence data using the NCBI database (https://blast.ncbi.nlm.nih.gov/Blast.cgi, accessed on 3 February 2021). For *Fusarium* isolates, we additionally analyzed the sequence homology of the elongation factor-1 alpha (EF-1a) gene sequencing and performed blast searches using the fusarium MLST database (http://www.cbs.knaw.nl/fusarium/, accessed on 6 February 2019) [13]. For *Aspergillus* isolates, the homology of the β-tubulin genes was analyzed to identify the species. Based on the results of the morphology and sequence homology, we comprehensively performed species identification.

## 3. Results 

### 3.1. Characteristics of IFD Cases

Table 1 shows the characteristics of the IFD cases with haematological malignancies identified in our centre between 2013 and 2019. Many cases were difficult to treat, as was identifying their causative fungi, and most of the requests to the MMRC were for identification and MIC measurement. The median age of IFD patients was 57.5 years (0–84 years). The HSCT patients were significantly younger than the non-HSCT patients. The most common underlying disease was acute myeloid leukaemia (AML) (34.3%), followed by acute lymphoblastic leukaemia (ALL) (17.0%) and malignant lymphoma (ML) (11.8%). 46.1% of the IFD patients received autologous or allogeneic HSCT (Table 2). In this analysis, 53.9% of the IFD patients were non-HSCT patients. Surprisingly, the incidence of IFD was higher in the non-HSCT patients than in the HSCT patients in this analysis. AML was the most common underlying disease in the non-HSCT patients, accounting for 24 cases. Most of the non-HSCT patients had relapsed/refractory acute leukaemia or ML requiring intensive chemotherapy, or severe MDS, which leads to severe neutropenia. These results showed that neutrophils play an essential role in preventing the development of IFD. The clinical specimens of the IFD cases are shown in the Figure. The most common sources of the IFD isolates were blood, followed by sputum/ bronchoalveolar lavage fluid (BALF). *Aspergillus* spp. and Mucorales were mainly identified from the sputum/BALF specimens, while *Candida*, *Trichosporon*, and *Fusarium* were mainly isolated from the blood specimens (Figure 2). 

### 3.2. Causative Species of IFD

Table 3 shows the causative fungi of IFD (genus level) in the patients with haematological disorders in Japan according to their HSCT status. The most common causative agent of IFD in both the HSCT and the non-HSCT groups was *Aspergillus*, as previously reported [4,14], followed by *Fusarium* in the HSCT group and *Candida* in the non-HSCT group. Especially in the allogeneic HSCT group, significant numbers of uncommon fungi such as *Fusarium*, *Trichosporon*, and Mucorales have been identified. Of the 26 causative organisms identified in this study, more than half, 14, were rare fungi with only one case each.

Table 4 shows the causative species of IFD in the genera *Aspergillus*, *Candida*, and *Fusarium*, respectively. Among the 23 causative species of *Aspergillus*, *A. fumigatus* was the most common with 10 cases. In addition, cryptic species such as *A. udagawae* and *A. viridinutans* were also detected. These species often have different antifungal susceptibility patterns and are difficult to identify morphologically. In the causative species of *Candida*, the most common species was *C. fermentati* with four cases, followed by *C. glabrata*, *C. palmioleophila*, *C. parapsilosis*, and *C. guilliermondii* with two cases each. *Candida albicans*, which is generally considered the most common, was found in only one case in this study. Thirteen *Fusarium* strains were identified, with *F. petroliphilum* being the most common with five cases, followed by *F. keratoplasticum* and *F. solani sensu stricto* with two cases each, as we previously reported [10]. 

Finally, focusing on the underlying diseases of each fungus, AML was the most common disease in all the *Aspergillus*, *Candida*, and *Fusarium* groups. There was no significant difference in the underlying diseases among these groups, but ML was found in a certain number of cases in the *Aspergillus* and *Candida* groups, but not in the *Fusarium* group.

## 4. Discussion

Because of the difficulty in identifying fungi and the rarity of IFD, many of the similar reports of IFD analysis had only a few cases of proven fungal infection. This is the first report to elucidate the causative organisms of IFD in patients with haematological diseases in Japan. In many cases, analysis of the causative organism may significantly impact the choice of antifungal agents. Therefore, our study highlighted the clinical significance of identifying the organism in as much detail as possible. Despite advances in antifungal treatments, IFD has increased in patients with haematological disorders, and especially in HSCT patients [15]. As the prognosis for IFD is poor, it is essential to identify the causative organism and begin treatment expeditiously. However, identification of the causative fungus is often complicated and sometimes impossible. For this reason, there have been very few reports on the detailed species of IFD in patients with haematological disorders, and none in Japan. Some reports from other countries have shown that *Aspergillus*, *Candida*, and *Fusarium* are common causative organisms of IFD in patients with haematological disorders [5,16], but many other species remain unclear. Our centre is commissioned to analyze about 200 cases of fungi per year from various regions in Japan and provide support to clinicians. Therefore, many clinical isolates are stored and maintained in the MMRC along with clinical information. This study analyzed the causative species and background of IFD in Japanese patients with haematological diseases using clinical information attached to the clinical strains stored in the MMRC database.

Our epidemiological study of IFD revealed that acute leukaemia was the most common underlying disease of IFD in Japan and that 46.1% of all IFD patients received HSCT. Patients with acute leukaemia or those who received HSCT are immunocompromised and are presumed to have a high incidence of IFD. On the other hand, the incidence of IFD in non-HSCT patients is also high and requires attention. In particular, AML was most common in the non-HSCT group, suggesting that long-term impairment of neutrophil function is the most significant risk factor for IFD.

We demonstrated that *Aspergillus*, *Candida*, and *Fusarium* were the primary pathogens (49% of total) in IFD among the haematological disorders in Japan, in harmony with previous reports [5,16], followed by *Cunninghamella*, *Scedosporium*, *Cryptococcus*, *Rhizopus*, and *Schizophyllum* (Table 3). Thus, it is crucial to focus on these species as causative agents. In this study, *A. fumigatus* was the most common *Aspergillus* species and *C. fermentati* and *F. petroliphilum* were the most common *Candida* and *Fusarium* species, respectively. Of the 26 causative organisms identified in this study, more than half, 14, were rare fungi with only one case each. Cryptic species such as *A. udagawae* and *A. viridinutans*, which often have different antifungal susceptibility patterns and are difficult to identify morphologically, were also detected, suggesting that accurate identification of the causative species is important clinically. *C. albicans* was found in only one case in this study. Because many patients with haematological diseases received prophylactic antifungals such as azoles, it is inferred that non-*albicans Candida* infection was common as a breakthrough infection. Furthermore, most of the fungal isolates sent to the MMRC are difficult to identify in a general laboratory, and therefore the possibility of selection bias must be considered. When it comes to *Fusarium*, *F. petroliphilum* was the most frequent, as we previously reported [10,11]. We showed in the previous study that the infectious source of invasive fusariosis (IF) is via hospital drain outlets. IF is frequently a fatal disease, as there are few antifungals for its treatment, making the prevention of IF crucial. In this study, the HSCT group has the second-largest amount of *Fusarium* after *Aspergillus*, and the necessity of cleaning the drain outlets of the hospitals to prevent infection must be emphasized.

Non-*albicans Candida* often has a different spectrum of drug susceptibility than *C. albicans*, and *Fusarium* is naturally resistant to many antifungal drugs. Therefore, immediate identification of the fungus and its drug susceptibility is crucial. In many cases, analysis of the causative organism may have a significant impact on the choice of antifungal agents. Therefore, our study indicated that it is clinically significant to identify the organism in as much detail as possible.

There are some limitations to this study. The strains analyzed in this study were often difficult to identify at general institutions. Therefore, the number of the most common fungi such as *C. albicans* does not reflect the real clinical practice. In addition, because we used clinical information attached with the strains in this study, we could not perform a detailed analysis of IFD onset, survival rate, and antifungal prophylaxis. Further studies are required to investigate the prognosis of IFD in Japan. However, no other reports have analyzed the causative species of IFD in such detail. We believe that our results will assist clinicians in their treatment of patients with IFD.

In conclusion, we investigated the epidemiology and causative species of IFD in patients with haematological disorders in Japan. We propose that rapid and accurate identification of the causative fungi of IFD is clinically essential.

## Figures and Tables

**Figure 1 jof-07-00806-f001:**
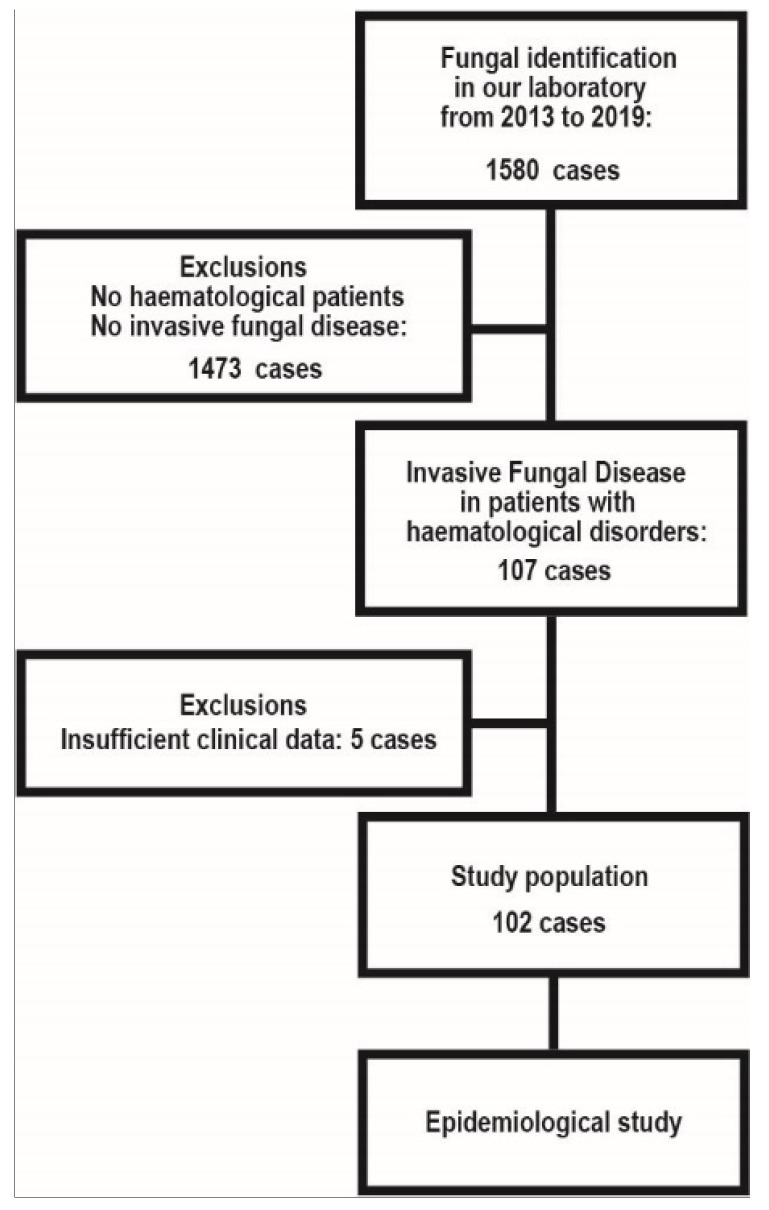
Study population and exclusion.

**Figure 2 jof-07-00806-f002:**
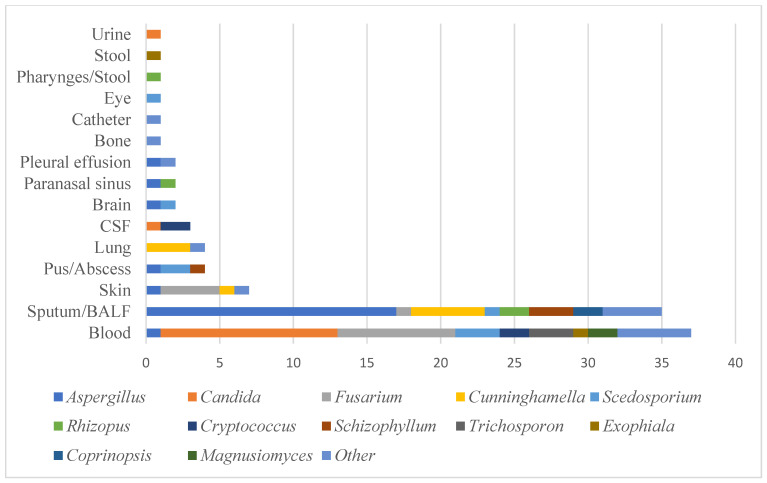
Clinical specimens in this study.

**Table 1 jof-07-00806-t001:** Characteristics of 102 IFD cases.

Characteristics	No. (%)
Gender	
Male	67 (65.7)
Female	35 (34.3)
Age yrs, median (range)	57.5 (0–84)
Underlying diseases	
Acute myeloid leukemia (AML)	35 (34.3)
Acute lymphoblastic leukemia (ALL)	18 (17.6)
Malignant lymphoma (ML)	12 (11.8)
Myelodysplastic syndrome (MDS)	11 (10.8)
Adult T-cell leukemia-lymphoma (ATL)	5 (4.9)
Multiple myeloma (MM)	5 (4.9)
Chronic granulomatous disease (CGD)	4 (3.9)
Chronic active EBV infection (CAEBV)	2 (1.9)
Chronic myeloid leukemia (CML)	2 (1.9)
Idiopathic thrombocytopenic purpura (ITP)	2 (1.9)
Primary myelofibrosis (PMF)	2 (1.9)
Waldenstrom Macroglobulinemia (WM)	2 (1.9)
Aplastic anemia (AA)	1 (1.0)
Hemophagocytic lymphohistiocytosis (HLH)	1 (1.0)

**Table 2 jof-07-00806-t002:** Patient characteristics: underlying diseases and transplant status.

Underlying Diseases	Non-HSCT	Allogeneic HSCT	Autologous SCT
AML	24	11	0
ALL	7	11	0
ML	10	2	0
MDS	5	6	0
ATL	3	2	0
MM	0	1	4
CGD	1	3	0
CAEBV	1	1	0
CML	1	1	0
ITP	2	0	0
PMF	0	2	0
WM	1	1	0
AA	0	1	0
HLH	0	1	0
Total	55	43	4

**Table 3 jof-07-00806-t003:** Causative fungi of IFD in patients with haematological disorders (genus level).

Genus	Non-HSCT	HSCT	Total
*Aspergillus*	12	11	23
*Candida*	9	5	14
*Fusarium*	6	7	13
*Cunninghamella*	5	4	9
*Scedosporium*	6	2	8
*Rhizopus*	1	3	4
*Schizophyllum*	2	2	4
*Cryptococcus*	3	1	4
*Trichosporon*	1	2	3
*Exophiala*	1	1	2
*Coprinopsis*	1	1	2
*Magnusiomyces*	2		2
*Scopulariopsis*		1	1
*Prototheca*		1	1
*Peroneutypa*		1	1
*Paecilomyces*		1	1
*Mucor*		1	1
*Trametes*	1		1
*Microsphaeropsis*	1		1
*Lomentospora*	1		1
*Isaria*	1		1
*Apiotrichum*	1		1
*Acremonium*	1		1
*Rhodotorula*		1	1
*Ochroconis*		1	1
*Gymnoascus*		1	1

**Table 4 jof-07-00806-t004:** Causative species of *Aspergillus*, *Candida*, and *Fusarium*.

Species	No.
* **Aspergillus** *	
*Aspergillus fumigatus*	10
*Aspergillus flavus*	2
*Aspergillus terreus*	2
*Aspergillus viridinutans*	2
*Aspergillus udagawae*	2
*Aspergillus creber*	1
*Aspergillus niger*	1
*Aspergillus siamensis*	1
*Aspergillus tubingensis*	1
*Aspergillus unguis*	1
* **Candida** *	
*Candida fermentati*	4
*Candida glabrata*	2
*Candida palmioleophila*	2
*Candida parapsilosis*	2
*Candida guilliermondii*	2
*Candida albicans*	1
*Candida orthopsilosis*	1
* **Fusarium** *	
*Fusarium petroliphilum*	5
*Fusarium keratoplasticum*	2
*Fusarium solani sensu stricto*	2
*Fusarium fujikuroi*	1
*Fusarium equiseti*	1
*Fusarium falciforme*	1
*Fusarium solani* (species complex)	1

## Data Availability

Not applicable.

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
