# Peer review of "A Reference Laboratory Surveillance on Fungal Isolates from Patients with Haematological Malignancy in Japan"

_jof, 2021, doi:10.3390/jof7100806_

Round 1
Reviewer 1 Report
This study aimed to investigate the epidemiology and causative organism of IFD in patients with haematological diseases in Japan. Unfortunately, we don't have information about outcome and treatment, antifungal prophylaxis and hospitalisation condition. Knowledge of epidemiology can help clinician to initiate treatment prior to identification.Author Response
This study aimed to investigate the epidemiology and causative organism of IFD in patients with haematological diseases in Japan. Unfortunately, we don't have information about outcome and treatment, antifungal prophylaxis and hospitalisation condition. Knowledge of epidemiology can help clinician to initiate treatment prior to identification.
Thank you for your comment. This study is not prospective: the fungal isolates used in this study are the ones that were sent from physicians in various medical facilities in Japan for requesting species identification and antifungal susceptibility testing. We performed this study based on the clinical information attached with the isolates. We don’t have the information about the outcome, treatment or antifungal prophylaxis, and hospitalisation condition. We completely agree with the comment and will plan a prospective study in the future.
Reviewer 2 Report
The authors provided a laboratory surveillance study of fungal isolates sent to a reference central laboratory from hematological patients with IFD in Japan. Overall the study is easy to follow. Below are some comments on its technical merits.
In my opinion, the title may seem somewhat overstated. This is not a study about IFD in Japan, but a laboratory surveillance on isolates from IFD in Japan. The provided clinical data is limited (for example, besides demographics and underlying disease, how many patients have proven vs. probable IFD? Other comorbidities? Antifungal prophylaxis? Which?)
If the authors intend to provide a more detailed epidemiological report, information on the prevalence (or, better if time-person data is available, incidence rate) of IFD in the overall hematological population should be added. In addition, information on patients with probable IFD but without positive culture (i.e., patients with positive fungal antigens) could be important for providing a clear epidemiological picture
It could be of interest to describe the heterogeneity of distribution of the different genus and species across the different laboratories that sent the isolates.
Some isolates were excluded according to the methods. A flow-chart of isolates inclusion could be added as a figure with precise numbers and types of excluded isolates.
Author Response
The authors provided a laboratory surveillance study of fungal isolates sent to a reference central laboratory from hematological patients with IFD in Japan. Overall the study is easy to follow. Below are some comments on its technical merits.
In my opinion, the title may seem somewhat overstated. This is not a study about IFD in Japan, but a laboratory surveillance on isolates from IFD in Japan. The provided clinical data is limited (for example, besides demographics and underlying disease, how many patients have proven vs. probable IFD? Other comorbidities? Antifungal prophylaxis? Which?)
As your helpful comments, we changed the title to “A reference laboratory surveillance on fungal isolates from patients with haematological malignancy in Japan.”
If the authors intend to provide a more detailed epidemiological report, information on the prevalence (or, better if time-person data is available, incidence rate) of IFD in the overall hematological population should be added. In addition, information on patients with probable IFD but without positive culture (i.e., patients with positive fungal antigens) could be important for providing a clear epidemiological picture
We completely agree with the comment. As you mentioned, data on the prevalence of IFD will improve our study. However, this study is not prospective: the fungal isolates used in this study are the ones that were sent from physicians in various medical facilities in Japan for requesting species identification and antifungal susceptibility testing. We do not have detailed information about the prevalence of IFD so far and will plan a prospective study in the future.
It could be of interest to describe the heterogeneity of distribution of the different genus and species across the different laboratories that sent the isolates.
Thank you for your fruitful comments. The strains analyzed in this study were provided by 60 medical institutions in 24 prefectures nationwide. Most medical institutions provided only a few isolates, making it difficult to compare the heterogeneity of distribution of different genera and species among the laboratories in this study.
Some isolates were excluded according to the methods. A flow-chart of isolates inclusion could be added as a figure with precise numbers and types of excluded isolates.
Thank you for your advice. As your helpful comment, we added the flow-chart (Figure1) and the Figure Legend.